# Online parent training platform for complementary treatment of disruptive behavior disorders in attention deficit hyperactivity disorder: A randomized controlled trial protocol

**Gabrielle Chequer de Castro Paiva**[1☯]*, **Daniel Augusto Ferreira e Santos**[1‡], **Julia Silva Jales**[2‡], **Marco Aurélio Romano-Silva**[3‡], **Débora Marques de Miranda**[4☯]

**1** Molecular Medicine Postgraduate Program, Faculty of Medicine, Federal University of Minas Gerais, Belo Horizonte, Minas Gerais, Brazil, **2** Research Center of Impulsivity and Attention, Faculty of Medicine, Federal University of Minas Gerais, Belo Horizonte, Minas Gerais, Brazil, **3** Department of Mental Health, Faculty of Medicine, Federal University of Minas Gerais, Belo Horizonte, Minas Gerais, Brazil, **4** Department of Pediatrics, Faculty of Medicine, Federal University of Minas Gerais, Belo Horizonte, Minas Gerais, Brazil

☯ These authors contributed equally to this work.
‡ DAFS, JSJ and MARS also contributed equally to this work.
* gabriellechequer@gmail.com

## Abstract

### Background

Attention Deficit/Hyperactivity Disorder (ADHD) is associated with a diversity of impairments and Oppositional Defiant Disorder (ODD) is a very frequent comorbidity. Parent Training, as an evidence-based intervention, seems effective in reducing externalizing/disruptive behaviors, possibly leading to a better prognosis. This clinical trial aims to evaluate the effectiveness of an online parent training model as a complementary treatment for ADHD and ODD.

### Methods

Patients and their families will be screened upon their entry into the Research Center of Impulsivity and Attention (NITIDA) at UFMG—Brazil. Ninety families whose children are male, between 6–12 years old, and have significant externalizing symptoms and whose primary caregiver have complete high school education will be invited to participate. Families will be randomized (1:1) into 03 groups: 1) standard care; 2) standard care + face-to-face parent training; 3) standard care + online parent training. Interventions are analogous, differing only in delivery format. In the face-to-face format, the intervention will be conducted by a specialized therapist and the online format will be carried out through a platform. There will be six sessions/modules, arranged on a weekly basis. Measures of externalizing symptoms, parental and children quality of life, parental stress and parenting style will be collected at baseline and after the intervention.

**Data Availability Statement:** No datasets were generated or analysed during the current study. When the research is completed, deidentified data will be made to be available. Address: Av. Presidente Antônio Carlos,6627 2˚ Ad Sl 2005. Unidade Administrativa II. Postal code: 31.270-901. Belo Horizonte. Minas Gerais. Brasil. Phone: +55 (31)3409-4592. E-mail: coep@prpq.ufmg.br.

**Funding:** . The funding sources were Coordination for the Improvement of Higher Education Personnel (CAPES); Research Program for the SUS (PPSUS: Foundation for Research Support of the State of Minas Gerais; Secretary of State for Health of Minas Gerais; Ministry of Health Brazil); and National Council for Scientific and Technological Development (CNPq). The funders had no role in study design, data collection and analysis, decision to publish, or preparation of the manuscript.

**Competing interests:** No. The authors have declared that no competing interests exist.

## Discussion

This clinical trial intends to verify the effects of a new, online, model of an evidence-based intervention, which would allow a wider access in the Brazilian context.

## Trial registration

Registered on Brazilian Registry of Clinical Trials (ReBEC). Number: RBR-6cvc85. July 24th (2020) 05:35 pm.

## Introduction

### Background and rationale

**Background and rationale.** Attention Deficit/Hyperactivity Disorder (ADHD) is one of the most common neurodevelopmental disorders in childhood, affecting more than 5% of the world's child population [1]. The disorder is characterized by symptoms of inattention-disorganization, hyperactivity and impulsivity, in addition to high clinical heterogeneity in terms of the course of symptoms and functional outcomes [2]. ADHD is associated with functional, social and academic impairments and the need for treatment increases for each additional symptom [3]. ADHD is a chronic disorder of development and should not be seen only as a disorder that affects children's behavior and learning [4]. The cumulative impairment of ADHD throughout life is considerable, including a higher mortality rate plus an increased risk of suicide, social adversity, high rate of comorbidities with others mental disorders (~ 84%) such as substance abuse, anxiety and mood disorders, personality disorders and disruptive disorders [4, 5]. Compared to children with typical development, children with ADHD usually have greater social disadvantages (lower family income, lower education and higher dropout rates, etc.) [6]. The individual losses of ADHD also translate into public costs in the field of education (special education, interventions with occupational therapy, speech therapy and physical education, school retention, psychological counseling due to disciplinary incidents at school etc.) and health care (primary care, pharmacological treatment, emergency services, care for behavioral and emotional problems, etc.) [7]. Another public cost is related to judicial problems of ADHD adolescents, leading to institutionalization. For adults, the costs are associated with loss of income and productivity at work, which increases with ADHD severity, lower salaries compared to healthy adults, difficulty in finding and maintaining a job, higher rate of absenteeism at work due to health reasons, difficulties in relationships with colleagues, in addition to more use of health services and problems with the law due to criminal behavior [7–9].

Disruptive Behavior disorders are the most easily identifiable conditions among those coexisting with ADHD, as they involve notable behaviors such as "tantrums," physical aggression, excessive argumentation, theft and other forms of challenge and resistance to authority [10]. The disruptive disorder most often coexisting with ADHD is Oppositional Defiant Disorder (ODD), which can reach 50% of cases [10]. Children with externalizing behaviors are more sensitive to the effects of hyper-reactive parenting [11]. The most negative interaction about the child seems to be elicited by the inherent externalizing problems [12, 13]. Children with ADHD usually talk too much, are more distracted, demanding, sulky and uncooperative which is more challenging for parents and makes them more stressed [14]. Parental stress, in turn, contributes to the adoption of negative parenting strategies such as authoritarianism, as well as disruptive problems in children and inefficient implementation of interventions [15–

17]. The severity of behavioral and disruptive problems in children is closely associated, therefore, with parental problems, and the family environment may be a determining factor in the prognosis and presence of comorbidities [18]. These disruptive behaviors cause long-term damage, being associated with negative and poorly adaptive outcomes in adult life such as risky sexual behavior, worse academic performance and professional achievement, greater number of infractions and car accidents, in addition to the presence of comorbidities such as anxiety and substance abuse [18]. Effective interventions in childhood, resulting in the control of these behaviors, promote a significant reduction of these losses in adulthood [19]. In addition to drug treatment, Parent Training (PT) has long been considered extremely effective in reducing externalizing / disruptive behaviors [20, 21].

Among existing Parent Training models, the Kazdin method is widely used [21, 22]. It consists of 12 meetings, where social learning techniques are taught to parents to change their children's behavior and communication and integration patterns at home [22]. It is based on four pillars: conceptual vision on how to change problems, set of techniques, development of specific skills and treatment assessment [22]. The objectives of PT are: to increase knowledge about the causes of children's bad behavior, to teach parents to pay attention to children's behavior, also discovering its positive aspects, to improve the efficiency of parents' authority to deal with your children's bad behavior, increase child's cooperation, distend relationships and promote family harmony [22]. This program is entirely based on behavioral science. Ciesielski, Loren and Tamm [23] observed that a behavioral parent training program significantly reduced refusal behaviors, especially in relation to daily tasks, homework, meals and interaction with peers. In addition, the program was effective in reducing parental stress [23].

In a Cochrane summary, Trevedi observed an improvement in parents' mental health related to group parent training programs [24]. In addition, the intervention reduced symptoms of anxiety, depression, guilt and anger and perceived stress in parents. There was also an increase in satisfaction with the relationship with the partner and a sense of parental confidence. The study pointed to short-term improvement, showing the need for some type of maintenance [24]. A randomized clinical trial showed PT's effectiveness in reducing ADHD symptoms and family tension, and improving parental self-efficacy. The positive effects remained for 36 weeks after treatment and the findings were restricted to parental reporting [25]. A review study on parent training interventions in ADHD, Zwi and collaborators observed a positive effect on children's behavior and a significant reduction in parental stress, as pointed out in other studies [26]. Despite this, the review highlights the low methodological quality of the included studies. In another review, regarding treatment available and recommended for ADHD, psychosocial interventions still present inconsistent evidence, especially in reducing symptoms of ADHD [27]. The review draws attention to the need for new studies that combine pharmacological intervention with other types of treatment, with a lack in the literature, especially concerning adolescent groups [27].

Several factors are associated with heterogeneity in the clinical response to PT. Among these are severe psychiatric comorbidities in patients with ADHD, and severe psychiatric conditions diagnosed in primary caregivers, in addition to severe social adversity in the home context, such as deprivation of basic rights [22]. Low socioeconomic status is strongly related to negative outcomes in children with ADHD, such as low school performance [28] and higher frequency of Oppositional Defiant Disorder (ODD) as a comorbidity [29]. Despite inconsistent findings, several studies have promoted the effectiveness of PT [23]. However, PT is a high-cost intervention and, thus, not commonly available in the public sphere in low-middle income countries such as Brazil. With a high prevalence, ADHD and ODD affect a large number of children whose families have low socioeconomic status, requiring a viable and low-cost alternative.

Some recent studies evaluated the feasibility of Parent Training implemented through online platforms. In a systematic review on the PT in digital format, several programs were evaluated for their effectiveness, considering that four out of seven programs were originally taught in person and were adapted to the digital format [30]. Of the studies that reported positive behavioral outcomes, the average effect size for parents' results was d = 0.46 and for results related to children's behavior, it was d = 0.61 [30]. Rabbitt and colleagues compared two groups of parents, whose children had conduct problems, who went through PT, one undergoing interventions exclusively in person and the other with very little contact between parents and therapists [31]. In both cases, the children showed a significant improvement in terms of functionality, adaptation and reduction of externalizing problems. The study points out, however, that parents reported better satisfaction with the module entirely in person, despite similar effects in all post-treatment assessments [31]. Högström and his collaborators also evaluated the effectiveness of a PT program administered online [32]. The results show a significant decrease in children's disruptive behaviors, over 18 months, since the beginning of the intervention that lasted 10 weeks. However, parenting skills seemed to worsen over time, demonstrating the need for some type of training maintenance [32]. A meta-analysis study demonstrated that the digital format of parent training promotes significant improvement in children's behavior when compared to those without treatment [33]. The study also pointed out that interactive computer programs were more effective than non-interactive ones [33]. Another review study demonstrated that the digital format, with reduced professional support, has similar evidence to the traditional format of parent training [34]. In addition, when compared to an untreated group, self-directed programs achieved significant improvements in the child's behavior (children <9 years, d = 0.47–0.80, 4 studies; and adolescents d = 0.17, 0.20, 2 studies) [34]. Preliminary indicators also suggested that improvements in technology may increase the involvement and results of standard treatment [34].

Considering the functional impact that disruptive behaviors add to ADHD, we will investigate the feasibility and effectiveness of a fully virtual parent training program when compared to face-to-face PT. Moreover, to assess the importance of this treatment, outcomes of intervention groups will be compared with a naive third group of children. The present study protocol has the potential to develop an accessible intervention technology aimed at families, of low/medium socioeconomic level, of children with serious externalizing problems. Socioeconomic level and organization of family environment are moderators of ADHD symptoms, demonstrating the importance of the interaction between genetics and environment in the clinical presentation/severity of the disorder [35]. This relationship can also occur due to the difficulty of access to health services, faced by low-income families. The trial, therefore, can reach vulnerable and unassisted populations. Although PT is costly in economic and human resources, it seems to have important and positive effects on the development of children [23].

**Choice of comparators.** Parent training as proposed by Kazdin is originally delivered in face-to-face format, being a well-consolidated and evidence-based program [22]. Thus, the online format is a new proposal and its effectiveness in relation to the traditional format should be investigated. Likewise, the potential benefit of an online PT in relation to conventional treatment, which would not include this type of intervention, should be explored, since the program can bring benefits regardless of the effectiveness in relation to the traditional delivery format. All patients will be under psychiatric follow-up at the Research Center of Impulsivity and Attention (NITIDA) and therefore will be receiving conventional treatment, which especially includes pharmacological treatment, at medical discretion, according to the clinical protocol. One group will receive only the usual care, and the other two will be submitted to PT, one in online format and the other in face-to-face format.

## Objectives

The objectives of this randomized clinical trial are: Investigate whether a behavioral parent training program, adapted from the model proposed by Alan Kazdin [22], is effective as a complementary treatment for ADHD and ODD, reducing externalizing symptoms; Investigate the effects of two modalities of parent training (online and face-to-face) on 1) externalizing symptoms reduction, 2) on childrens' and parents' quality of life, 3) on parenting style and 4) on stress perceived by the main caregiver, by comparing results before and after the intervention, in each of the modalities and between the modalities, checking out if the effects are similar; Evaluate the effects of parent training in both modalities (online and face-to-face) in short term; Evaluate if there is a moderating effect of the parents' intelligence, schooling, socioeconomic level, and symptoms of depression, anxiety and ADHD, on intervention response, in each of the parent training modalities (online and face-to-face); Evaluate if there is a moderating effect of the child's age and school performance on intervention response, in each of the parent training modalities (online and face-to-face).

## Trial design

The PT trial is designed as a randomized, controlled, experimental, open, single center, equivalence trial, with three-arm parallel groups. Randomization will be performed as block randomization with a 1:1 allocation.

## Methods: Participants, interventions, outcomes

### Study setting

All groups will be invited to the Research Center of Impulsivity and Attention (NITIDA) at Hospital das Clínicas, UFMG–Brazil.

### Eligibility criteria

Participants inclusion criteria: Families whose children have Attention Deficit/Hyperactivity Disorder and/or Oppositional Defiant Disorder; children must have externalizing symptoms such as hyperactivity and/or oppositional behavior; children who are between 6 and 12 years old; male gender. Primary caregiver with complete high school education.

Participants exclusion criteria: Families whose children have severe psychiatric comorbidities (i.e. psychosis, bipolar affective disorder, severe depression), genetic or neurological; whose intelligence is below the 5th percentile; feminine gender. Families whose primary caregivers have intelligence below the 5th percentile or have been diagnosed with serious psychiatric conditions (i.e. psychoses, bipolar affective disorder, severe depression); and families with severe social adversity in the home context (i.e. domestic violence, physical abuse, extreme poverty, hunger). Primary caregiver without complete high school education.

All parents must sign the free and informed consent form, as well as the children the assent form. The terms will be delivered during the first consultation of the participants in the outpatient clinic, if they wish to participate voluntarily in the research. If there is more than one primary caregiver, it is recommended that everyone participate in the intervention. The effective participation of caregivers will be reported and analyzed along with the results.

### Interventions

**Interventions.** Participants will be randomly assigned to either control or one of the experimental groups with a 1:1 allocation as per a computer-generated randomization schedule stratified by children's age using five permuted blocks of 6 participants.

**GROUP 01:** 30 participants. Conventional treatment, without complementary treatment. Bi-monthly consultations with a child and adolescent psychiatrist, including drug treatment if there is a clinical indication (at medical discretion, according to the clinical protocol), and without any other type of complementary treatment.

**GROUP 02:** 30 participants. Conventional treatment and behavioral intervention in face-to-face format. Bi-monthly consultations with a child and adolescent psychiatrist, including drug treatment if there is a clinical indication (at medical discretion, according to the clinical protocol), and complementary behavioral intervention: parent management training, in face-to-face format, with a specialized therapist, in six sessions held on a weekly basis, adapted from the "Parent Management Training" manual developed by Alan Kazdin [22].

**GROUP 03:** 30 participants. Conventional treatment and behavioral intervention in online format. Bi-monthly consultations with a child and adolescent psychiatrist, including drug treatment if there is a clinical indication (at medical discretion, according to the clinical protocol), and complementary behavioral intervention: parent management training, in online format, on a platform developed for the study, in six modules, to be carried out on a weekly basis, adapted from the "Parent Management Training" manual developed by Alan Kazdin [22].

The parent training will be carried out following the model proposed by Kazdin [22], adapted from 12 to 6 sessions (more extensive), and excluding those stages where the child's presence was requested (see Fig 1), in order to make this project feasible for low-income

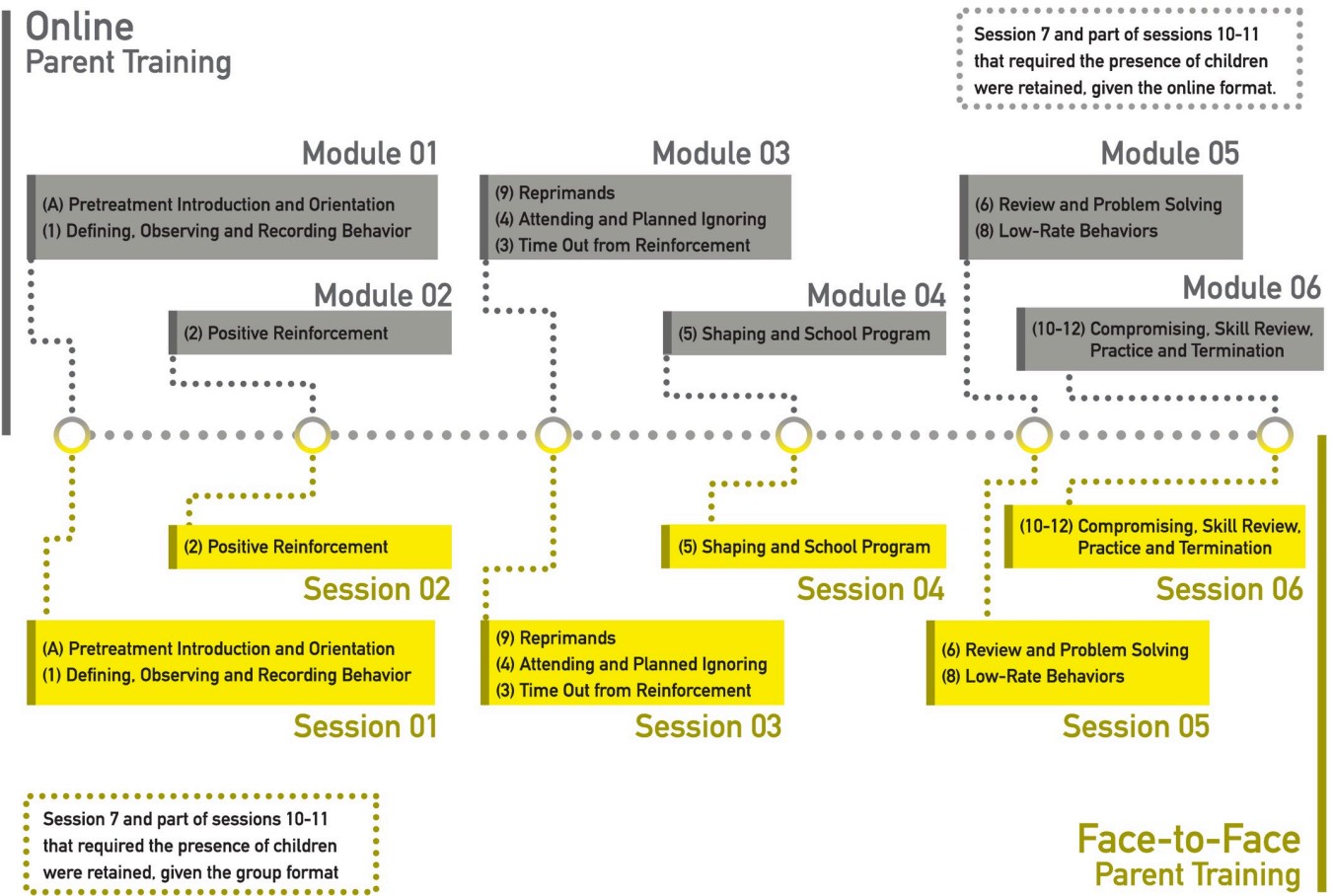

**Fig 1. Parent training sessions/modules.** Intervention adapted from the "Parent Manegement Training" manual developed by Alan Kazdin (2005).

families. Face-to-face version will be carried out individually in a weekly-based 6 sessions guided by a specialized therapist. PT online will have 6 modules, whose objectives will be analogous to those of face-to-face sessions. In online version, parents will have access to videos in animations or recordings, that must be seen sequentially and considering one module per week. Parents should watch all videos available for each module, with an approximate duration of 30 minutes in total, and they may be watched more than once. At the end of each module, a summary booklet will be made available for download, in addition to other necessary support materials according to the topics covered in the module (a sheet for recording the behavior of the child, for example). In online version, parents will have no contact with the therapist and they should answer some questions about what was addressed in that module, reaching 60% of correct answers to access the next module. In face-to-face version, the summary booklet and others support materials made available will be the same as the online format.

The face-to-face intervention will be carried out by psychologists from the Research Center of Impulsivity and Attention (NITIDA), previously trained and with at least five years of experience in behavioral psychology.

**Modifications.**    The participant will be excluded from the survey if he does not attend or does not watch regularly the weekly sessions of the intervention. If the patient reports the impossibility of attending the session and is available to reschedule it within a maximum of 30 days, or in the case of the online group, watch the content up to a maximum of 30 days after the scheduled date, the patient can proceed with the intervention. The intervention does not offer any type of risk to the patient, and therefore discontinuity is only related to attendance.

Participants whose diagnosis on screening differs from the diagnosis of the psychiatry team will be excluded from the study.

**Adherence.**    The use of the platform is monitored, and the authors are notified at each stage completed by the participant. One day before the appropriate date for the execution of each module (a week apart between them) the platform automatically sends an e-mail, with the intention of reminding the participant of the commitment to the intervention and already anticipating a little of what will be seen in the next module (teaser). In addition, telephone contacts, when made available by the participant, enable direct contact if a delay in the deadlines is identified. Likewise, in the case of face-to-face meetings, the patient will be contacted frequently.

**Concomitant care.**    Treatment as usual will be delivered for all patients, including drug treatment if there is a clinical indication (at medical discretion, according to the clinical protocol). The treatment can also include speech therapy and occupational therapy, if there is a clinical indication. Other behavioral interventions concurrently will not be allowed.

Although the psychiatric consultations offered by Research Center of Impulsivity and Attention (NITIDA) start at the same time as the first assessment before behavioral intervention, some children may eventually already be on medication or other non-pharmacological treatment. The study will report these treatments, but only participants with another behavioral intervention at the same time will be excluded.

## Outcomes

Primary outcomes: Change in the symptoms of Attention Deficit/Hyperactivity Disorder and/ or Oppositional Defiant Disorder are expected, with a decrease in externalizing symptoms, determined by the Multimodal Treatment Assessment Study—Swanson, Nolan e Pelham (MTA-SNAP-IV) scale: which assesses symptoms of attention deficit/hyperactivity disorder and Oppositional Defiant Disorder in children and adolescents [36]; and by the Kiddie Schedule for Affective Disorders and Schizophrenia for School Aged Children—Lifetime Version

(K-SADS-PL, 2013) interview: semi-structured interview that raises important information about the history of psychiatric disorders, at the present time and throughout life and assesses the severity of symptoms [37]. The quantity and intensity of symptoms will be checked and data will be collected before the start of the intervention and again just after the end of the intervention. The MTA-SNAP-IV scale was adapted and has data from the studied clinical group in the Brazilian context [36]. The K-SADS-PL interview is based on the criteria of the most recent version of the Diagnostic and Statistical Manual of Mental Disorders DSMV [10].

Secondary outcomes: Change in parental style are expected with a tendency to democratic style, determined through the Parenting Styles and Dimensions Questionnaire—PSDQ (42 items version): Questionnaire that assesses how parents manage their relationship with their children and the child's behavior, considering the control dimensions and affection, being based on a theoretical model that categorizes parenting styles into four possible ones: democratic, authoritarian, permissive and negligent. This questionnaire was previously adapted and validated in the Brazilian context [38]. Furthermore, a decrease in the level of perceived stress by caregivers is expected, measured by the Perceived Stress Scale: measures how much individuals perceive situations as stressful. It is a Likert-type scale and contains 14 questions [39], this scale is the most used instrument to assess the perception of stress, having been validated in more than 20 countries, including Brazil [40]. Finally, is expected an improvement in the quality of life of parents and children, determined through two scales: World Health Organization Quality of Life abbreviated measure (WHOQOL-BREF): assesses quality of life. Instrument developed by the World Health Organization, in a version reduced from 100 to 26 items [41] and Kidscreen-52—Version of caregivers' report and self-report: Evaluates the quality of life of children and adolescents [42]. All scales will be filled before the start of the intervention and again just after the end of the intervention.

## Participant timeline

The participants timeline is shown below, in Fig 2.

## Sample size

A sample of 90 participants would be sufficient to detect a small (d = 0.2) to high (0.8) association with a small (0.4) covariance between repeated measures (intraclass correlation coefficient —ICC = 0.9) and an association of moderate (0.5) to high for larger ICCs (0.6 to 0.9).

## Recruitment

All groups will be invited to the Research Center of Impulsivity and Attention (NITIDA) at Hospital das Clínicas, UFMG–Brazil. NITIDA offers psychiatric outpatient care for children and adolescents with externalizing problems, in particular ADHD and ODD cases and being recognized by these services. On average, six new patients are screened per week and this project screening will continue until 100% of the expected sample is reached.

## Methods: Assignment of interventions

### Allocation

**Sequence generation.** Participants will be randomly assigned to either control or one of the experimental groups with a 1:1 allocation as per a computer-generated randomization schedule. The allocation sequence will be generated applying a permuted block design with random blocks, stratified by children's age. Throughout the study, the randomization will be

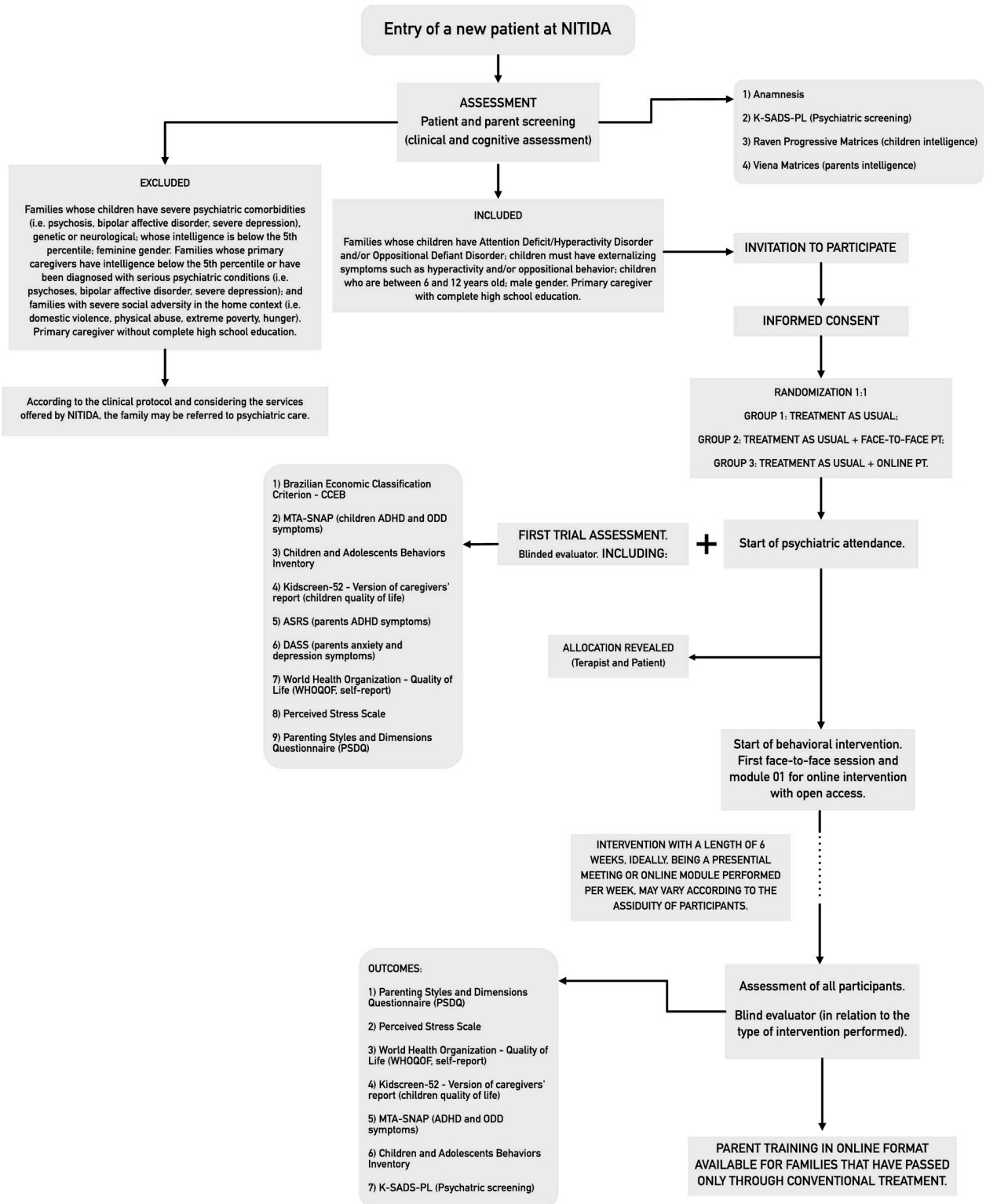

**Fig 2. Participant timeline.**

conducted by the computer technician without any influence of the principal investigators, raters or therapists.

**Concealment mechanism.** Allocation concealment will be ensured using sequentially numbered, opaque, sealed envelopes. The randomization code will be known by participants and therapists after all baseline measurements have been completed by the evaluator. The allocation will be revealed to the therapist and the participant the moment intervention is about to begin.

**Implementation.** All patients who give consent for participation and who fulfil the inclusion criteria will be randomized. Randomization will be requested by the staff member responsible for recruitment and clinical interviews and will be performed by the computer technician, both members of NITIDA. In return, the computer technician will send an answer form to the study therapist who is not involved in assessing the outcome of the study. This form will include a randomization number. Closed envelopes with printed randomization numbers on it will be available for the therapist. For every randomization number the corresponding code for the therapy group of the randomization list will be found inside the envelopes. The therapist will open the envelope and will find the treatment condition to be conducted in this patient. The therapist then gives the information about treatment allocation to the patient. Staff responsible for recruitment and symptom ratings is not allowed to receive information about the group allocation.

## Blinding

**Blinding.** Assessments will be conducted by a professional blind to treatment allocation. The assessor will go through an assessment training program with a specialist in the field. Due to the nature of the intervention neither participants nor the therapists can be blinded to allocation. Regardless of the allocation, the scales will be filled out by all participants on a tablet and the database will be fed automatically. The interviews and the cognitive measures collected by the appraiser will be passed on by himself to this database, which does not contain any information about the allocation and therefore it can be passed on to researchers.

Data management, statistician and staff responsible for recruitment and assessment will be blind to the type of intervention performed in each participant.

**Emergency unblinding.** The survey does not offer any risk to the participants, and therefore unblindings would not be applicable.

## Schedule of enrolment, interventions, and assessments

In Fig 3 the schedule of enrolment, interventions, and assessments can be seen.

## Methods: Data collection, management, analysis

### Data collection methods

**Data collection methods.** An initial screening will be carried out by the outpatient team, trained in evaluation procedures by a specialist. In this screening, general data important to the research, such as the parents' psychiatric history, for example, will be collected through an extensive anamnesis. In addition, the child's psychiatric screening will be carried out through the K-SADS-PL, an interview based entirely on the diagnostic criteria of the latest edition of the DSM [10, 37]. Intelligence tests (matrices) will be applied to the child and the primary caregiver, both validated and with Brazilian normative data [43, 44].

After the screening, the included participants will undergo a first trial assessment, including a battery of scales that will be completed in the presence of an evaluator, on a notebook. The

| | STUDY PERIOD | | | | | | |
|---|---|---|---|---|---|---|---|
| | **Enrolment** | **Allocation** | **Post-allocation** | | | | **Close-out** |
| **TIMEPOINT**** | **-$t_1$** | **0** | **$t_1$** | **$t_2$** | **$t_3$** | **$t_4$** | **$t_x$** |
| **ENROLMENT:** | | | | | | | |
| **Eligibility screen** | X | | | | | | |
| **Informed consent/assent** | X | | | | | | |
| **Allocation** | | X | | | | | |
| **INTERVENTIONS:** | | | | | | | |
| *Psychiatric Attendance* | | | X | X | X | X | |
| *Parent Training (online or face-to-face) or waiting time* | | | | X | | | |
| *Start online parent training (for waiting group)* | | | | | | X | |
| **ASSESSMENTS:** | | | | | | | |
| *First Trial Assessment* | | | X | | | | |
| *Outcomes Assessment* | | | | | X | | X |

**Fig 3. Schedule of enrolment, interventions, and assessments.**

scales are self-reported (parents' behavior; items 1, 2, 3, 4 and 5) or refer to the child's behavior, according to the parents' report (items 6, 7 and 8), besides the Brazilian Economic Classification Criterion [45].

1. ASRS—Adult Self-Report Scale (parents ADHD symptoms) [46].

2. Depression, Anxiety and Stress Scale (DASS) [47].

3. Perceived Stress Scale [48, 49].

4. Parenting Styles and Dimensions Questionnaire (PSDQ) [38].

5. World Health Organization Quality of Life abbreviated measure [41, 50].

6. MTA-SNAP (children ADHD and ODD symptoms) [51].

7. The Child and Adolescent Behavior Inventory (CABI) is a questionnaire useful for the preparation of screening for clinical evaluation [52].

8. Kidscreen-52 to evaluate the quality of life of children [42].

After the participant completes the intervention, or the waiting time (six weeks, for the control group), the K-SADS will be reapplied and a battery of scales containing the same instruments (with the exception of the ASRS and DASS) must be completed by the main caregiver.

The interval between the pre-assessment and the beginning of the intervention, as well as the end of the intervention and the execution of the post-assessment, will be as small as possible, and cannot exceed a maximum period of 60 days. The deadline was considered for project feasibility, taking into account the need for flexibility in scheduling between the team and parents.

SNAP and ASRS check ADHD symptoms in children and adults, respectively, including ODD in children. Both instruments were translated into Portuguese with a methodology of translation, back-translation, evaluation of semantic equivalence, polling the target population and choice of final version [36, 46]. The SNAP validity and reliability were evaluated in a Brazilian study, in a sample of children with ADHD, which demonstrated good psychometric properties in this context [51]. In the same direction, ASRS-18 showed good psychometric properties [46], and the study showed that the instrument can be used as a complementary measure for the diagnosis of ADHD in adults. The Depression, Anxiety and Stress Scale (DASS) was adapted and validated for use in a Brazilian context [47], as well as the Perceived Stress Scale [48, 49] and the Parenting Styles and Dimensions Questionnaire (PSDQ) [38]. The Child and Adolescent Behavior Inventory (CABI) is a questionnaire designed to collect information from the parents of children and adolescents, being useful for the preparation of screening for clinical evaluation [52]. World Health Organization Quality of Life abbreviated measure showed a good performance concerning internal consistency, discriminant validity, criterion validity, concurrent validity and test-retest reliability, been an interesting option to evaluate quality of life in Brazil [41, 50]. Kidscreen-52 was chosen to evaluate the quality of life of children, specifically. The translation, cross-cultural adaptation and psychometric qualities of the KIDSCREEN-52 questionnaire were satisfactory, which makes its application in the Brazilian population feasible [42]. The Brazil Economic Classification Criterion or CCEB aims to be a unique way of assessing the purchasing power of consumer groups, dividing the market into economy classes. The instrument was built using statistical techniques that are based on collective data [45].

**Retention.** The intervention has the potential to bring great benefits to the whole family and parents will be encouraged weekly through teasers received via email and WhatsApp to complete the process. Moreover, a report containing the child's intelligence and behavior assessment will be given to the parents. In addition, all children will be under medical supervision at the outpatient clinic (NITIDA) and doctors also recommend the continuity and completion of the intervention as a complementary treatment. Access to families is easy and if parents leave the PT, all post-intervention measures will be collected as soon as the dropout is confirmed (according to the treatment interruption criterion).

## Data management

A part of the data will be entered electronically. The intelligence scores and clinical interview results will be entered by the evaluator that are blind to intervention groups. Range checks for data values will be performed to promote data quality.

Participant files are to be stored in numerical order and stored in a secure and accessible place and manner. Participant files will be maintained in storage for a period of 10 years after completion of the study.

### Statistical methods

**Outcomes.** The main analysis strategies will involve techniques for group comparison (ANOVA etc.), analysis of repeated measures (mixed models, GHG etc.), as well as correlation and regression to verify the association between variables. Techniques will be chosen depending on data distribution. If necessary, depending on the sampling power, we will employ techniques resampling.

**Additional analyses.** The main additional analyses strategies will involve Anova post hoc tests. The subgroup analysis and adjusted analysis will depend on the preliminary data, such as depending on differences in parents or children IQ, socioeconomic variables and ADHD subgroups.

**Analysis population and missing data.** Participants who do not complete the intervention or the information about the outcomes will be excluded from the study and who delay for 30 days to participate on the PT.

## Methods: Monitoring

### Data monitoring

**Formal committee.** Data monitoring committee (DMC) is not needed, taking into account a short duration of the trial and known minimal risks. However, the diagnosis is confirmed by an expert team that communicates any diagnosis divergence.

**Interim analyses.** Individuals that will not fill the information of the primary outcomes will be excluded. Socioeconomic data will be analyzed to evaluate any potential of sampling bias. The clinical psychiatric assessment will be done by an experienced clinical team, any diagnosis divergence will be informed. If adherence is poor, the treatment will be early stopped. Adherence will be considered poor if the parent delay for more than 30 days to engage in the next session/module.

### Auditing

Not applicable. The diagnosis divergence will be evaluated and informed on a weekly basis.

## Ethics and dissemination

### Research ethics approval

Approved by Research Ethics Committee of the Federal University of Minas Gerais. CAAE: 98623218.2.0000.5149. Written, informed consent to participate will be obtained from all parents, as well as informed assent from the children.

### Protocol amendments

Any modifications to the protocol which may impact on the conduct of the study, potential benefit of the patient or may affect patient safety, will lead to a formal amendment to the protocol with REBEC, the ethics coittee and clinical trial publication journal.

### Consent or assent

All parents must sign the free and informed consent form, as well as the children the assent form. The terms will be delivered during the first consultation of the participants in the outpatient clinic, if they wish to participate voluntarily in the research.

## Confidentiality

All study-related information will be stored securely at the study site. All participant information will be stored in locked file cabinets in areas with limited access. All data collection forms will be identified by a coded ID [identification] number only to maintain participant confidentiality. All local databases will be secured with password-protected access systems.

## Dissemination policy

**Trial results.**   All research data and personal information will be under responsibility of the researchers in order to protect confidentiality before, during and after the trial. All parents or guardians' results will be communicated at the end and regarding the trial results.

Trial results will be published at REBEC, regardless of the magnitude or direction of effect. The results will also be reported in an original article and submitted for a relevant journal.

**Reproducible research.**   The anonymous data information might be available under request and reasonable demand.

## Discussion

In Brazil, 74.9% of households have internet access, and 97% of these users use cell phones or tablets as a means of access, in addition to desktops or laptops [53]. Digital interventions are a strategy to overcome the barriers of the face-to-face modality, in addition to saving time and expenses by reducing the time spent with travel, planning, with the professional's time and with the implementation of the in-person intervention [30]. Kazdin highlights the positive potential of these new forms of treatment, especially considering their reach to underserved families [54].

The use of technology and digital delivery is a growing parental training method, with high potential for reaching and sustainability, maximizing training fidelity, and engagement [30]. Engaging means engaging, and for this to really happen, the user needs to find, in addition to an attractive tool, a satisfying experience and fluid navigation, with design being an intersection point between technology and the transmission of knowledge, orchestrating and organizing resources to make this platform functional [55].

An online parent training platform was developed, which is a new (in the Brazilian context) and more accessible intervention technology aimed at low/medium income families with children who have externalizing problems. Low socioeconomic status is strongly related to negative outcomes in children with ADHD, such as poor school performance [28] and higher frequency of ODD as a comorbidity [29].

Kazdin mentions a gap in the distribution of treatment: there is a large difference in the proportion of people who suffer from a disorder in relation to the proportion of those who receive or are reached by treatment [54]. Currently, the vast majority of children and adolescents who need mental health services do not receive treatment and, although there are many barriers that prevent these services and these people from meeting, the largest of them is the dominant model for implementing interventions, which include face-to-face meetings with a qualified mental health professional, usually offered in a clinical setting (clinic, private practice, health center) [54]. Noting that this model greatly limits the scale and reach of these interventions, it highlights the use of technology to fill this gap, including recommending the use of computer network (internet)-based applications, such as social networks and discussion forums, and, consequently, distance learning platforms [54].

Interventions in the digital model, self-directed and with little or no contact with professionals, have been shown to be effective, corroborating the popularization and increase of internet access in all social classes [30, 53]. The use of the internet has changed the way the

general population has access to information, and it is used in many ways for numerous purposes, and the modern technology of knowledge dissemination has had a great impact on the health area, allowing patients to be reached without physical distance limit anywhere in the world, potentiating at a global level the exchange of information and data collection that enable the monitoring and improvement of health services [56].

Currently, behavioral interventions were implemented in the online model, using systems that require little or no intervention from a professional, and the main way these treatments are currently disseminated is through a website, usually based on evidence-based treatments [57]. Its effectiveness has been demonstrated by a large number of trials with mental health problems such as depression, anxiety, substance abuse and insomnia, as well as promising results in the treatment of bipolar mood disorder and schizophrenia [57]. There is a wide range of digitally well-established treatments for depression and anxiety, most of which are designed to be self-directed or to receive some form of support [58]. Most of these treatments are forms of cognitive-behavioral therapy, derived from face-to-face models, and generally have a strong educational bias. Some even present themselves as educational programs rather than treatments, and implement the intervention in "lessons", not "sessions" [57].

The platform was built using the Kazdin manual as a reference, however, in an adapted and not analogous way [22]. The sessions in which the child's presence was necessary were removed from the program, and others with similar themes grouped together, with the intention of reducing dropout rates. Kazdin discusses that the first six sessions would be significantly related to positive outcomes [22]. This adaptation was made with the aim that the material favors parents' learning, with an underlying theory that produces quick effects, considering that dropout rates increase with more sessions [30].

## Supporting information

**S1 File. SPIRIT checklist.**
(PDF)

**S2 File. Informed consent materials.**
(DOCX)

**S3 File. Administrative Information.**
(DOCX)

**S1 Protocol.**
(PDF)

## Acknowledgments

Acknowledgments to the entire Research Center of Impulsivity and Attention (NITIDA) team, including psychiatrists and residents, scientific initiation students and hospital staff. We especially acknowledge the NITIDA graphic designer and IT technician, Daniel Augusto Ferreira e Santos, who was responsible for the visual identity and development of the online parent training platform.

## Author Contributions

**Conceptualization:** Gabrielle Chequer de Castro Paiva, Daniel Augusto Ferreira e Santos, Débora Marques de Miranda.

**Funding acquisition:** Débora Marques de Miranda.

**Investigation:** Gabrielle Chequer de Castro Paiva.

**Methodology:** Gabrielle Chequer de Castro Paiva, Daniel Augusto Ferreira e Santos, Débora Marques de Miranda.

**Project administration:** Gabrielle Chequer de Castro Paiva, Daniel Augusto Ferreira e Santos, Marco Aurélio Romano-Silva, Débora Marques de Miranda.

**Resources:** Débora Marques de Miranda.

**Supervision:** Gabrielle Chequer de Castro Paiva, Débora Marques de Miranda.

**Writing – original draft:** Gabrielle Chequer de Castro Paiva.

**Writing – review & editing:** Gabrielle Chequer de Castro Paiva, Daniel Augusto Ferreira e Santos, Julia Silva Jales, Marco Aurélio Romano-Silva, Débora Marques de Miranda.

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
