## [Decision Letter · Decision Letter 0]

13 Apr 2022

PONE-D-21-38630Online Parent Training Platform for complementary treatment of Disruptive Behavior Disorders in Attention Deficit Hyperactivity Disorder: a randomized controlled trial protocolPLOS ONE

Dear Dr. Chequer de Castro Paiva,

Thank you for submitting your manuscript to PLOS ONE. After careful consideration, we feel that it has merit but does not fully meet PLOS ONE’s publication criteria as it currently stands. Therefore, we invite you to submit a revised version of the manuscript that addresses the points raised during the review process.

We look forward to receiving your revised manuscript.

Kind regards,

Walid Kamal Abdelbasset, Ph.D.

Academic Editor

PLOS ONE

Journal Requirements:

2. We note that you have stated that you will provide repository information for your data at acceptance. Should your manuscript be accepted for publication, we will hold it until you provide the relevant accession numbers or DOIs necessary to access your data. If you wish to make changes to your Data Availability statement, please describe these changes in your cover letter and we will update your Data Availability statement to reflect the information you provide."

Reviewers' comments:

Reviewer's Responses to Questions

**Comments to the Author**

1. Does the manuscript provide a valid rationale for the proposed study, with clearly identified and justified research questions?

Reviewer #1: Yes

Reviewer #2: Yes

2. Is the protocol technically sound and planned in a manner that will lead to a meaningful outcome and allow testing the stated hypotheses?

Reviewer #1: Yes

Reviewer #2: Yes

3. Is the methodology feasible and described in sufficient detail to allow the work to be replicable?

Reviewer #1: Yes

Reviewer #2: Yes

4. Have the authors described where all data underlying the findings will be made available when the study is complete?

Reviewer #1: Yes

Reviewer #2: Yes

5. Is the manuscript presented in an intelligible fashion and written in standard English?

Reviewer #1: Yes

Reviewer #2: Yes

6. Review Comments to the Author

You may also provide optional suggestions and comments to authors that they might find helpful in planning their study.

Reviewer #1: A very interesting piece of experiment. The topic is a subject which has a great need for studies and trials and I am thrilled to see this trial. The ADHD treatment is quite complex and requires ongoing monitoring and the current available treatment protocols are not obviously the most desired treatment plans. Therefore observing the new treatment and alternative plans is a breath of fresh air. Thanks

Reviewer #2: The topic is really interesting to investigate especially in a country like Brazil where utilization of low cost healthcare services in underserved communities is a main need. Yet, I have some questions and comments regarding the methodology:

1)Why did you chose male gender only?

2)Would it be possible for more than one caregiver of the same patient to join the study?

3)The possibility of watching online videos more than once might be a potential bias since patients in the face to face group won't have the same option to revisit the material.

4)why didn't you do the post-session knowledge assessment to both face to face and online groups?

5) Will the effect of pharmacotherapy regarding dosing and type of medications be controlled during randomization of patients as it might be a major confounding factor regarding the patient outcomes.

7. PLOS authors have the option to publish the peer review history of their article (what does this mean?). If published, this will include your full peer review and any attached files.

Reviewer #1: **Yes: **Dr Lily Abedipour MD

Reviewer #2: No

---

## [Author Response · Author response to Decision Letter 0]

17 May 2022

Dear Editor and Reviewers, 

Many thanks for your careful revision. 

Regarding journal requirements, we have re-reviewed the list of references and PLOS ONE's style requirements. About repository information for our data, the current manuscript does not have preliminary data. The results of the study will be published once the research is completed and deidentified data will be made to be available under reasonable request to the correspondent author. We added this information to the cover letter.

Below are answers to the questions raised by Reviewer #2

Why did you choose male gender only?

In order to have a representative sample of the studied population, since the prevalence is much higher in boys and our number of participants is relatively small. Also, the clinical presentation of ADHD in girls can be very different (Hinshaw, 2018).

Hinshaw SP. Attention Deficit Hyperactivity Disorder (ADHD): Controversy, Developmental Mechanisms, and Multiple Levels of Analysis. Annu Rev Clin Psychol. 2018 May 7;14:291-316. doi: 10.1146/annurev-clinpsy-050817-084917. Epub 2017 Dec 8. PMID: 29220204.

2)Would it be possible for more than one caregiver of the same patient to join the study?

Yes, it is even recommended that all primary caregivers participate (Kazdin,2005). In the pre-intervention evaluation, there is a question that seeks to investigate the caregivers' intention to participate and in the post-intervention evaluation, a question that seeks to quantify the size of the actual participation of each caregiver. 

A change has been made to the text with the intention of clarifying this issue. The change can be found in last paragraph of the "Eligibility criteria" session: 

“If there is more than one primary caregiver, it is recommended that everyone participate in the intervention. The effective participation of caregivers will be reported and analyzed along with the results.”

3)The possibility of watching online videos more than once might be a potential bias since patients in the face-to-face group won't have the same option to revisit the material.

A fundamental difference from the face-to-face group is the frequent contact with the therapist, who helps parents in the application and not only in the exposure of the content, so we understand that each of the groups has specific favorable aspects inherent to the delivery format of intervention. This can be considered an important strategy to transform a face-to-face intervention into a fully self-directed model. This change must necessarily have materials and strategies to facilitate learning (Breitenstein et al., 2014).

Breitenstein, S. M., Gross, D., & Christophersen, R. (2014). Digital delivery methods of parenting training interventions: a systematic review. Worldviews on Evidence-Based Nursing / Sigma Theta Tau International, Honor Society of Nursing, 11(3), 168–176.

4) Why didn't you do the post-session knowledge assessment to both face to face and online groups?

We understand that the evaluation after each session in the face-to-face group was not necessary since the feedback of the learned content is immediate during the session, including the therapist's considerations and interventions to verify that the content is clear and that the participant has no doubts about its application to the case. The idea of the test after the online modules would be precisely to fill the information gap and to rise potential frails.

5) Will the effect of pharmacotherapy regarding dosing and type of medications be controlled during randomization of patients as it might be a major confounding factor regarding the patient outcomes.

Pharmacological treatment will be conducted by the medical team, following standard protocol and concomitantly with parent training. Frequency of drug treatment and medium dose will be described along with the results and specific statistical analyzes can be performed according to the results. The intent of the study is to verify the effectiveness of parent training as a complementary treatment to standard care.

---

## [Decision Letter · Decision Letter 1]

21 Jul 2022

Online Parent Training Platform for complementary treatment of Disruptive Behavior Disorders in Attention Deficit Hyperactivity Disorder: a randomized controlled trial protocol

PONE-D-21-38630R1

Dear Dr. Chequer de Castro Paiva,

We’re pleased to inform you that your manuscript has been judged scientifically suitable for publication and will be formally accepted for publication once it meets all outstanding technical requirements.

Kind regards,

Walid Kamal Abdelbasset, Ph.D.

Academic Editor

PLOS ONE

Additional Editor Comments (optional):

Reviewers' comments:

Reviewer's Responses to Questions

**Comments to the Author**

1. Does the manuscript provide a valid rationale for the proposed study, with clearly identified and justified research questions?

Reviewer #2: Yes

2. Is the protocol technically sound and planned in a manner that will lead to a meaningful outcome and allow testing the stated hypotheses?

Reviewer #2: Yes

3. Is the methodology feasible and described in sufficient detail to allow the work to be replicable?

Reviewer #2: Yes

4. Have the authors described where all data underlying the findings will be made available when the study is complete?

Reviewer #2: Yes

5. Is the manuscript presented in an intelligible fashion and written in standard English?

Reviewer #2: Yes

6. Review Comments to the Author

You may also provide optional suggestions and comments to authors that they might find helpful in planning their study.

Reviewer #2: All comments from first review were answered properly and minor edit was done accordingly so the manuscript is ready to proceed to next level

7. PLOS authors have the option to publish the peer review history of their article (what does this mean?). If published, this will include your full peer review and any attached files.

Reviewer #2: **Yes: **Ahmed Abdelkarim

---

## [Editor Report · Acceptance letter]

11 Aug 2022

PONE-D-21-38630R1 

Online Parent Training Platform for complementary treatment of Disruptive Behavior Disorders in Attention Deficit Hyperactivity Disorder: a randomized controlled trial protocol 

Dear Dr. Chequer de Castro Paiva:

I'm pleased to inform you that your manuscript has been deemed suitable for publication in PLOS ONE. Congratulations! Your manuscript is now with our production department. 

Kind regards, 

on behalf of

Dr. Walid Kamal Abdelbasset 

Academic Editor

PLOS ONE